# BC-IRL: Learning Generalizable Reward Functions from Demonstrations

**Andrew Szot**[1,2], **Amy Zhang**[1], **Dhruv Batra**[1,2], **Zsolt Kira**[2], **Franziska Meier**[1]
[1]Meta AI, [2]Georgia Tech

## Abstract

How well do reward functions learned with inverse reinforcement learning (IRL) generalize? We illustrate that state-of-the-art IRL algorithms, which maximize a maximum-entropy objective, learn rewards that overfit to the demonstrations. Such rewards struggle to provide meaningful rewards for states not covered by the demonstrations, a major detriment when using the reward to learn policies in new situations. We introduce BC-IRL, a new inverse reinforcement learning method that learns reward functions that generalize better when compared to maximum-entropy IRL approaches. In contrast to the MaxEnt framework, which learns to maximize rewards around demonstrations, BC-IRL updates reward parameters such that the policy trained with the new reward matches the expert demonstrations better. We show that BC-IRL learns rewards that generalize better on an illustrative simple task and two continuous robotic control tasks, achieving over twice the success rate of baselines in challenging generalization settings.

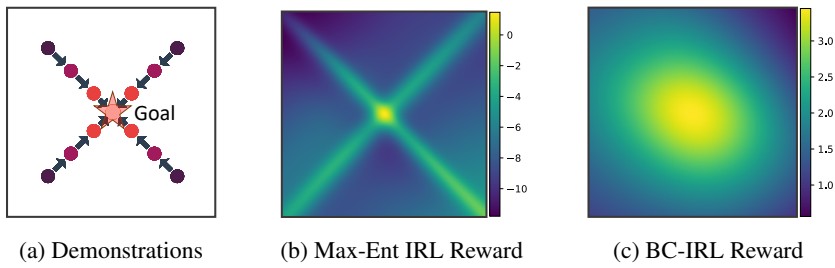

| (a) Demonstrations | (b) Max-Ent IRL Reward | (c) BC-IRL Reward |

Figure 1: A visualization of learned rewards on a task where a 2D agent must navigate to the goal at the center. Figure 1a: Four trajectories are provided as demonstrations and the demonstrated states are visualized as points. Rewards learned via Maximum Entropy are in Figure 1b and BC-IRL in Figure 1c. Lighter colors represent larger predicted rewards. The MaxEnt objective overfits to the demonstrations, giving high rewards only close to the expert states, preventing the reward from providing meaningful learning signals in new states.

## 1 Introduction

Reinforcement learning has demonstrated success on a broad range of tasks from navigation Wijmans et al. (2019), locomotion Kumar et al. (2021); Iscen et al. (2018), and manipulation Kalashnikov et al. (2018). However, this success depends on specifying an accurate and informative reward signal to guide the agent towards solving the task. For instance, imagine designing a reward function for a robot window cleaning task. The reward should tell the robot how to grasp the cleaning rag, how to use the rag to clean the window, and to wipe hard enough to remove dirt, but not hard enough to break the window. Manually shaping such reward functions is difficult, non-intuitive, and time-consuming. Furthermore, the need for an expert to design a reward function for every new skill limits the ability of agents to autonomously acquire new skills.

Inverse reinforcement learning (IRL) (Abbeel & Ng, 2004; Ziebart et al., 2008; Osa et al., 2018) is one way of addressing the challenge of acquiring rewards by learning reward functions from demonstrations and then using the learned rewards to learn policies via reinforcement learning. When compared to direct imitation learning, which learns policies from demonstrations directly, potential benefits of IRL are at least two-fold: first, IRL does not suffer from the compounding error problem that is often observed with policies directly learned from demonstrations (Ross et al., 2011; Barde et al., 2020); and second, a reward function could be a more abstract and parsimonious description of

the observed task that generalizes better to unseen task settings (Ng et al., 2000; Osa et al., 2018). This second potential benefit is appealing as it allows the agent to learn a reward function to train policies not only for the demonstrated task setting (e.g. specific start-goal configurations in a reaching task) but also for unseen settings (e.g. unseen start-goal configurations), autonomously without additional expert supervision.

However, thus far the generalization properties of reward functions learned via IRL are poorly understood. Here, we study the generalization of learned reward functions and find that prior IRL methods fail to learn generalizable rewards and instead overfit to the demonstrations. Figure 1 demonstrates this on a task where a point mass agent must navigate in a 2D space to a goal location at the center. An important reward characteristic for this task is that an agent, located anywhere in the state-space, should receive increasing rewards as it gets closer to the goal. Most recent prior work Fu et al. (2017); Ni et al. (2020); Finn et al. (2016c) developed IRL algorithms that optimize the maximum entropy objective (Ziebart et al., 2008) (Figure 1b), which fails to capture goal distance in the reward. Instead, the MaxEnt objective leads to rewards that separate non-expert from expert behavior by maximizing reward values along the expert demonstration. While useful for imitating the experts, the MaxEnt objective prevents the IRL algorithms from learning to assign meaningful rewards to other parts of the state space, thus limiting generalization of the reward function.

As a remedy to the reward generalization challenge in the maximum entropy IRL framework, we propose a new IRL framework called **Behavioral Cloning Inverse Reinforcement Learning (BC-IRL)**. In contrast to the MaxEnt framework, which learns to maximize rewards around demonstrations, the BC-IRL framework updates reward parameters such that the policy trained with the new reward matches the expert demonstrations better. This is akin to the model-agnostic meta-learning (Finn et al., 2017) and loss learning (Bechtle et al., 2021) frameworks where model or loss function parameters are learned such that the downstream task performs well when utilizing the meta-learned parameters. By using gradient-based bi-level optimization Grefenstette et al. (2019), BC-IRL can optimize the behavior cloning loss to learn the reward, rather than a separation objective like the maximum entropy objective. Importantly, to learn the reward, BC-IRL differentiates through the reinforcement learning policy optimization, which incorporates exploration and requires the reward to provide a meaningful reward throughout the state space to guide the policy to better match the expert. We find BC-IRL learns more generalizable rewards (Figure 1c), and achieves over twice the success rate of baseline IRL methods in challenging generalization settings.

Our contributions are as follows: 1) The general BC-IRL framework for learning more generalizable rewards from demonstrations, and a specific BC-IRL-PPO variant that uses PPO as the RL algorithm. 2) A quantitative and qualitative analysis of reward functions learned with BC-IRL and Maximum-Entropy IRL variants on a simple task for easy analysis. 3) An evaluation of our novel BC-IRL algorithm on two continuous control tasks against state-of-the-art IRL and IL methods. Our method learns rewards that transfer better to novel task settings.

## 2 BACKGROUND AND RELATED WORK

We begin by reviewing Inverse Reinforcement Learning through the lens of bi-level optimization. We assume access to a rewardless Markov decision process (MDP) defined through the tuple $\mathcal{M} = (\mathcal{S}, \mathcal{A}, \mathcal{P}, \rho_0, \gamma, H)$ for state-space $\mathcal{S}$, action space $\mathcal{A}$, transition distribution $\mathcal{P}(s'|s, a)$, initial state distribution $\rho_0$, discounting factor $\gamma$, and episode horizon $H$. We also have access to a set of expert demonstration trajectories $\mathcal{D}^e = \{\tau_i^e\}_{i=1}^N$ where each trajectory is a sequence of state, action tuples.

IRL learns a parameterized reward function $R_\psi(\tau_i)$ which assigns a trajectory a scalar reward. Given the reward, a policy $\pi_\theta(a|s)$ is learned which maps from states to a distribution over actions. The goal of IRL is to produce a reward $R_\psi$, such that a policy trained to maximize the sum of (discounted) rewards under this reward function matches the behavior of the expert. This is captured through the following bi-level optimization problem:

$$\min_\psi \mathcal{L}_{\text{IRL}}(R_\psi; \pi_\theta) \qquad \textbf{(outer obj.)} \qquad (1a)$$

$$\text{s.t. } \theta \in \operatorname*{argmax}_\theta g(R_\psi, \theta) \qquad \textbf{(inner obj.)} \qquad (1b)$$

where $\mathcal{L}_{\text{IRL}}(R_\psi; \pi_\theta)$ denotes the IRL loss and measures the performance of the learned reward $R_\psi$ and policy $\pi_\theta$; $g(R_\psi, \theta)$ is the reinforcement learning objective used to optimize policy parameters $\theta$. Algorithms for this bi-level optimization consist of an outer loop ((1a)) that optimizes the reward and an inner loop ((1b)) that optimizes the policy given the current reward.

**Maximum Entropy IRL:** Early work on IRL learns rewards by separating non-expert from expert trajectories (Ng et al., 2000; Abbeel & Ng, 2004; Abbeel et al., 2010). A primary challenge of these early IRL algorithms was the ambiguous nature of learning reward functions from demonstrations: many possible policies exist for a given demonstration, and thus many possible rewards exist. The Maximum Entropy (MaxEnt) IRL framework (Ziebart et al., 2008) seeks to address this ambiguity, by learning a reward (and policy) that is as non-committal (uncertain) as possible, while still explaining the demonstrations. More concretely, given reward parameters $\psi$, MaxEnt IRL optimizes the log probability of the expert trajectories $\tau^e$ from demonstration dataset $\mathcal{D}^e$ through the following loss,

$$\mathcal{L}_{\text{MaxEnt-IRL}}(R_\psi) = -\mathbb{E}_{\tau^e \sim \mathcal{D}^e}\left[\log p(\tau^e|\psi)\right] = -\mathbb{E}_{\tau^e \sim \mathcal{D}^e}\left[\log \frac{\exp\left(R_\psi(\tau^e)\right)}{Z(\psi)}\right]$$

$$= -\mathbb{E}_{\tau^e \sim \mathcal{D}^e}\left[R_\psi(\tau^e)\right] + \log Z(\psi).$$

A key challenge of MaxEnt IRL is estimating the partition function $Z(\psi) = \int \exp R_\psi d\tau$. Ziebart et al. (2008) approximate $Z$ in small discrete state spaces with dynamic programming.

**MaxEnt from the Bi-Level perspective:** However, computing the partition functions becomes intractable for high-dimensional and continuous state spaces. Thus algorithms approximate $Z$ using samples from a policy optimized via the current reward. This results in the partition function estimate being a function of the current policy $\log \hat{Z}(\psi; \pi_\theta)$. As a result, MaxEnt approaches end up following the bi-level optimization template by iterating between: 1) updating reward function parameters given current policy samples via the outer objective ((1a)); and 2) optimizing the policy parameters with the current reward parameters via an inner policy optimization objective and algorithm (1b). For instance, model-based IRL methods such as Wulfmeier et al. (2017); Levine & Koltun (2012); Englert et al. (2017) use model-based RL (or optimal control) methods to optimize a policy (or trajectory), while model-free IRL methods such as Kalakrishnan et al. (2013); Boularias et al. (2011); Finn et al. (2016b;a) learn policies via model-free RL in the inner loop. All of these methods use policy rollouts to approximate either the partition function of the maximum-entropy IRL objective or its gradient with respect to reward parameters in various ways (outer loop). For instance Finn et al. (2016b) learn a stochastic policy $q(\tau)$, and sample from that to estimate $Z(\psi) \approx \frac{1}{M}\sum_{\tau_i \sim q(\tau)} \frac{\exp R_\psi(\tau_i)}{q(\tau_i)}$ with $M$ samples from $q(\tau)$. Fu et al. (2017) with adversarial IRL (AIRL) follow this idea and view the problem as an adversarial training process between policy $\pi_\theta(a|s)$ and discriminator $D(s) = \frac{\exp R_\psi(s)}{\exp R_\psi(s) + \pi_\theta(a|s)}$. Ni et al. (2020) analytically compute the gradient of the $f$-divergence between the expert state density and the MaxEnt state distribution, circumventing the need to directly compute the partition function.

**Meta-Learning and IRL:** Like some prior work (Xu et al., 2019; Yu et al., 2019; Wang et al., 2021; Gleave & Habryka, 2018; Seyed Ghasemipour et al., 2019), BC-IRL combines meta-learning and inverse reinforcement learning. However, these works focus on fast adaptation of reward functions to new tasks for MaxEnt IRL through meta-learning. These works require demonstrations of the new task to adapt the reward function. BC-IRL algorithm is a fundamentally new way to learn reward functions and does not require demonstrations for new test settings. Most related to our work is Das et al. (2020), which also uses gradient-based bi-level optimization to match the expert. However, this approach requires a pre-trained dynamics model. Our work generalizes this idea since BC-IRL can optimize general policies, allowing any objective that is a function of the policy and any differentiable RL algorithm. We show our method, without an accurate dynamics model, outperforms Das et al. (2020) and scales to more complex tasks where Das et al. (2020) fails to learn.

**Generalization in IRL:** Some prior works have explored how learned rewards can generalize to training policies in new situations. For instance, Fu et al. (2017) explored how rewards can generalize to training policies under changing dynamics. However, most prior work focuses on improving policy generalization to unseen task settings by addressing challenges introduced by the adversarial training objective of GAIL (Xu & Denil, 2019; Zolna et al., 2020; 2019; Lee et al., 2021; Barde et al., 2020; Jaegle et al., 2021; Dadashi et al., 2020). Finally, in contrast to most related work on generalization, our work focuses on analyzing and improving reward function transfer to new task settings.

## 3 LEARNING REWARDS VIA BEHAVIORAL CLONING INVERSE REINFORCEMENT LEARNING (BC-IRL)

We now present our algorithm for learning reward functions via behavioral cloning inverse reinforcement learning. We start by contrasting the maximum entropy and imitation loss objectives for

inverse reinforcement learning in Section 3.1. We then introduce a general formulation for BC-IRL in Section 3.2, and present an algorithmic instantiation that optimizes a BC objective to update the *reward* parameters via gradient-based bi-level optimization with a model-free RL algorithm in the inner loop in Section 3.3.

## 3.1 OUTER OBJECTIVES: MAX-ENT VS BEHAVIOR CLONING

In this work, we study an alternative IRL objective from the maximum entropy objective. While this maximum entropy IRL objective has led to impressive results, it is unclear how well this objective is suited for learning reward functions that generalize to new task settings, such as new start and goal distributions. Intuitively, assigning a high reward to demonstrated states (without task-specific hand-designed feature engineering) makes sense when you want to learn a reward function that can recover exactly the expert behavior, but it leads to reward landscapes that do not necessarily capture the essence of the task (e.g. to reach a goal, see Figure 1b).

Instead of specifying an IRL objective that is directly a function of reward parameters (like maximum entropy), we aim to measure the reward function's performance through the policy that results from optimizing the reward. With such an objective, we can optimize reward parameters for what we care about: for the resulting policy to match the behavior of the expert. The behavioral cloning (BC) loss measures how well the policy and expert actions match, defined for continuous actions as $\mathbb{E}_{(s_t,a_t)\sim\tau^e}\left(\pi_\theta(s_t) - a_t\right)^2$ where $\tau^e$ is an expert demonstration trajectory. Policy parameters $\theta$ are a result of using the current reward parameters $\psi$, which we can make explicit by making $\theta$ a function of $\psi$ in the objective: $\mathcal{L}_{\text{BC-IRL}} = \mathbb{E}_{(s_t,a_t)\sim\tau^e}(\pi_{\theta(\psi)}(s_t) - a_t)^2$. The IRL objective is now formulated in terms of the policy rollout "matching" the expert demonstration through the BC loss.

We use the chain-rule to decompose the gradient of $\mathcal{L}_{\text{BC-IRL}}$ with respect to reward parameters $\psi$. We also expand how the policy parameters $\theta(\psi)$ are updated via a REINFORCE update with learning rate $\alpha$ to optimize the current reward $R_\psi$ (but any differentiable policy update applies).

$$\frac{\partial}{\partial\psi}\mathcal{L}_{\text{BC-IRL}} = \frac{\partial}{\partial\psi}\left[\mathbb{E}_{(s_t,a_t)\sim\tau^e}\left[\left(\pi_{\theta(\psi)}(s_t) - a_t\right)^2\right]\right] = \mathbb{E}_{(s_t,a_t)\sim\tau^e}\left[2\left(\pi_{\theta(\psi)}(s_t) - a_t\right)\right]\frac{\partial}{\partial\psi}\pi_{\theta(\psi)}$$

$$\text{where } \theta(\psi) = \theta_{\text{old}} + \alpha \mathbb{E}_{(s_t,a_t)\sim\pi_{\theta_{\text{old}}}}\left[\left(\sum_{k=t+1}^T \gamma^{k-t-1}R_\psi(s_k)\right)\nabla\ln\pi_{\theta_{\text{old}}}(a_t|s_t)\right] \quad (2)$$

Computing the gradient for the reward update in Equation (2) includes samples from $\pi$ collected in the reinforcement learning (RL) inner loop. This means the reward is trained on diverse states beyond the expert demonstrations through data collected via exploration in RL. As the agent explores during training, BC-IRL must provide a meaningful reward signal throughout the state-space to guide the policy to better match the expert. Note that this is a fundamentally different reward update rule as compared to current state-of-the-art methods that maximize a maximum entropy objective. We show in our experiments that this results in twice as high success rates compared to state-of-the-art MaxEnt IRL baselines in challenging generalization settings, demonstrating that BC-IRL learns more generalizable rewards that provide meaningful rewards beyond the expert demonstrations.

The BC loss updates only the reward, as opposed to updating the policy as typical BC for imitation learning does Bain & Sammut (1995). BC-IRL is a IRL method that produces a reward, unlike regular BC that learns only a policy. Since BC-IRL uses RL, not BC, to update the policy, it avoids the pitfalls of BC for policy optimization such as compounding errors. Our experiments show that policies trained with rewards from BC-IRL generalize over twice as well to new settings as those trained with BC. In the following section, we show how to optimize this objective via bi-level optimization.

## 3.2 BC-IRL

We formulate the IRL problem as a gradient-based bi-level optimization problem, where the outer objective is optimized by differentiating through the optimization of the inner objective. We first describe how the policy is updated with a fixed reward, then how the reward is updated for the policy to better match the expert.

**Inner loop (policy optimization):** The inner loop optimizes policy parameters $\theta$ given current reward function $R_\psi$. The inner loop takes $K$ gradient steps to optimize the policy given the current reward. Since the reward update will differentiate through this policy update, we require the policy update to be differentiable with respect to the reward function parameters. Thus, any reinforcement learning algorithm which is differentiable with respect to the reward function parameters can be plugged in here, which is the case for many policy gradient and model-based methods. However, this does not

include value-based methods such as DDPG Lillicrap et al. (2015) or SAC Haarnoja et al. (2018) that directly optimize value estimates since the reward function is not directly used in the policy update.

**Outer loop (reward optimization)**: The outer loop optimization updates the reward parameters $\psi$ via gradient descent. More concretely: after the inner loop, we compute the gradient of the outer loop objective $\nabla_\psi \mathcal{L}_{\text{BC-IRL}}$ wrt to reward parameters $\psi$ by propagating through the inner loop. Intuitively, the new policy is a function of reward parameters since the old policy was updated to better maximize the reward. The gradient update on $\psi$ tries to adjust reward function parameters such that the policy trained with this reward produces trajectories that match the demonstrations more closely. We use Grefenstette et al. (2019) for this higher-order optimization.

---

**Algorithm 1** BC-IRL (general framework)

1: Initial reward $R_\psi$, policy $\pi_\theta$
2: Policy updater POLICY_OPT($R, \pi$)
3: Expert demonstrations $\mathcal{D}^e$
4: **for** each epoch **do**
5:     Policy Update:
6:     $\theta' \leftarrow \text{POLICY\_OPT}(R_\psi, \pi_\theta)$
7:     Sample demo batch $\tau^e \sim \mathcal{D}^e$
8:     Compute IRL loss
9:     $\mathcal{L}_{\text{BC-IRL}} = \mathbb{E}_{(s_t, a_t) \sim \tau^e} \left( \pi_{\theta'}(s_t) - a_t \right)^2$
10:    Compute gradient of IRL loss wrt reward
11:    $\nabla_\psi \mathcal{L}_{\text{BC-IRL}} = \frac{\partial \mathcal{L}_{\text{BC-IRL}}}{\partial \theta'} \frac{\partial \text{POLICY\_OPT}(R_\psi, \pi_\theta)}{\partial \psi}$
12:    $\psi \leftarrow \psi - \nabla_\psi \mathcal{L}_{\text{BC-IRL}}$
13: **end for**

---

BC-IRL is summarized in Algorithm 1. Line 5 describes the inner loop update, where we update the policy $\pi_\theta$ to maximize the current reward $R_\psi$. Lines 6-7 compute the BC loss between the updated policy $\pi_{\theta'}$ and expert actions sampled from expert dataset $\mathcal{D}^e$. The BC loss is then used in the outer loop to perform a gradient step on reward parameters in lines 8-9, where the gradient computation requires differentiating through the policy update in line 5.

### 3.3 BC-IRL-PPO

We now instantiate a specific version of the BC-IRL framework that uses proximal policy optimization (PPO) Schulman et al. (2017) to optimize the policy in the inner loop. This specific version, called BC-IRL-PPO, is summarized in Algorithm 2.

BC-IRL-PPO learns a state-only parameterized reward function $R_\psi(s)$, which assigns a state $s \in \mathcal{S}$ a scalar reward. The state-only reward has been shown to lead to rewards that generalize better Fu et al. (2017). BC-IRL-PPO begins by collecting a batch of rollouts in the environment from the current policy (line 5 of Algorithm 2). For each state $s$ in this batch we evaluate the learned reward function $R_\psi(s)$ (line 6). From this sequence of rewards, we compute the advantage estimates $\hat{A}_t$ for each state (line 7). As is typical in PPO, we also utilize a learned value function $V_\nu(s_t)$ to predict the value of the starting and ending state for partial episodes in the rollouts.

---

**Algorithm 2** BC-IRL-PPO

1: Initial reward $R_\psi$, policy $\pi_\theta$, value function $V_\nu$
2: Expert demonstrations $\mathcal{D}^e$
3: **for** each epoch **do**
4:     **for** $k = 1 \rightarrow K$ **do**
5:        Run policy $\pi_\theta$ in environment for $T$ timesteps
6:        Compute rewards $\hat{r}_t^\psi$ for rollout with current $R_\psi$
7:        Compute advantages $\hat{A}^\psi$ using $\hat{r}^\psi$ and $V_\nu$
8:        Compute $\mathcal{L}_{\text{PPO}}$ using $\hat{A}^\psi$
9:        Update $\pi_\theta$ with $\nabla_\theta \mathcal{L}_{\text{PPO}}$
10:    **end for**
11:    Sample demo batch $\tau^e \sim \mathcal{D}^e$
12:    Compute $\mathcal{L}_{\text{BC-IRL}} = \mathbb{E}_{(s_t, a_t) \sim \tau^e} \left( \pi_\theta(s_t) - a_t \right)^2$
13:    Update reward $R_\psi$ with $\nabla_\psi \mathcal{L}_{\text{BC-IRL}}$
14: **end for**

---

This learned value function $V_\nu$ is trained to predict the sum of future discounted rewards for the current reward function $R_\psi$ and policy $\pi_\theta$ (part of $\mathcal{L}_{\text{PPO}}$ in line 8). Using the advantages, we then compute the PPO update (line 9 of Algorithm 2) using the standard PPO loss in equation 8 of Schulman et al. (2017). Note the advantages are a function of the reward function parameters used to compute the rewards, so PPO is differentiable with respect to the reward function. Next, in the outer loop update, we update the reward parameters, by sampling a batch of demonstration transitions (line 11), computing the behavior cloning IRL objective $\mathcal{L}_{\text{BC-IRL}}$ (line 12), and updating the reward parameters $\psi$ via gradient descent on $\mathcal{L}_{\text{BC-IRL}}$ (line 13). Finally, in this work, we perform one policy optimization step ($K = 1$) per reward function update. Furthermore, rather than re-train a policy from scratch for every reward function iteration, we initialize each inner loop from the previous $\pi_\theta$. This initialization is important in more complex domains where $K$ would otherwise have to be large to acquire a good policy from scratch.

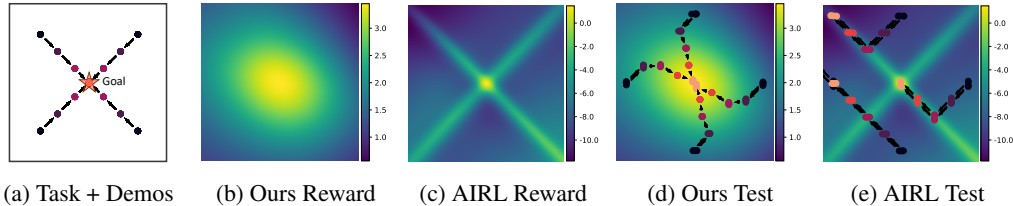

| (a) Task + Demos | (b) Ours Reward | (c) AIRL Reward | (d) Ours Test | (e) AIRL Test |

Figure 2: Results on the point mass navigation task. We show the learned reward functions of our method (b) vs. AIRL (c) and the policies learned from scratch using those reward functions (d, e).

## 4 ILLUSTRATION & QUALITATIVE ANALYSIS OF LEARNED REWARDS

We first analyze the rewards learned by different IRL methods in a 2D point mass navigation task. The purpose of this analysis is to test our hypothesis that our method learns more generalizable rewards compared to maximum entropy baselines in simple low-dimensional settings amenable to intuitive visualizations. Specifically, we compare BC-IRL-PPO to the following baselines.

**Exact MaxEntIRL (MaxEnt)** Ziebart et al. (2008): The exact MaxEntIRL method where the partition function is exactly computed by discretizing the state space.

**Guided Cost Learning (GCL)** Finn et al. (2016b): Uses the maximum-entropy objective to update the reward. The partition function is approximated via adaptive sampling.

**Adversarial IRL (AIRL)** Fu et al. (2017): An IRL method that uses a learned discriminator to distinguish expert and agent states. As described in Fu et al. (2017) we also use a shaping network $h$ during reward training, but only visualize and transfer the reward approximator $g$.

**f-IRL** Ni et al. (2021): Another MaxEntIRL based method, f-IRL computes the analytic gradient of the f-divergence between the agent and expert state distributions. We use the JS divergence version.

Our method does not require demonstrations at test time, instead we transfer our learned rewards zero-shot. Thus we forego comparisons to other meta-learning methods, such as Xu et al. (2019), which require test time demonstrations. While a direct comparison with Das et al. (2020) is not possible because their method assumes access to a pre-trained dynamics model, we conduct a separate study comparing their method with an oracle dynamics model against BC-IRL in Appendix A.5. All baselines use PPO Schulman et al. (2017) for policy optimization as commonly done in prior work Orsini et al. (2021). All methods learn a state-dependent reward $r_\psi(s)$, and a policy $\pi(s)$, both parametrized as neural networks. Further details are described in Appendix C.

The 2D point navigation tasks consist of a point agent policy that outputs a desired change in $(x, y)$ position (velocity) $(\Delta x, \Delta y)$ at every time step. The task has a trajectory length of $T = 5$ time steps with 4 demonstrations. Figure 2a visualizes the expert demonstrations where darker points are earlier time steps. The agent starting state distribution is centered around the starting state of each demonstration.

Figure 2b,c visualize the rewards learned by BC-IRL and the AIRL baseline. Lighter regions indicate higher rewards. In Figure 2b, BC-IRL learns a reward that looks like a quadratic bowl centered at the origin, which models the distance to the goal across the entire state space. AIRL, the maximum entropy baseline, visualized in Figure 2c, learns a reward function where high rewards are placed on the demonstrations and low rewards elsewhere. Other baselines are visualized in Appendix Figure 4.

To analyze the generalization capabilities of the learned rewards we use them to train policies on a new starting state distribution (visualized in Appendix Figure 9). Concretely, a newly initialized policy is trained from scratch to maximize the learned reward from the testing start state distribution. The policy is trained with 5 million environment steps, which is the same number of steps as for learning the reward. The testing starting state distribution has no overlap with the training start state distribution. Policy optimization at test time is also done with PPO. The Figure 2d,e display trajectories from the trained policies where darker points again correspond to earlier time steps.

This qualitative evaluation shows that BC-IRL learns a meaningful reward for states not covered by the demonstrations. Thus at test time agent trajectories are guided towards the goal with the terminal states (lightest points) close to the goal. The X-shaped rewards learned by the baselines do not provide meaningful rewards in the testing setting as they assign uniformly low rewards to states not covered by the demonstration. This provides poor reward shaping which prevents the agent from reaching the goal within the 5M training interactions with the environment. This results in agent trajectories that do not end close to the goal by the end of training.

|  | BC-IRL | AIRL | GCL | MaxEnt | f-IRL |
|---|---|---|---|---|---|
| **Train** | $0.03 \pm 0.01$ | $0.08 \pm 0.00$ | $0.00 \pm 0.00$ | NA | $0.07 \pm 0.01$ |
| **Eval (Train)** | $0.03 \pm 0.00$ | $0.08 \pm 0.02$ | $2.07 \pm 0.09$ | $0.30 \pm 0.40$ | $0.08 \pm 0.03$ |
| **Eval (Test)** | $\mathbf{0.04} \pm \mathbf{0.01}$ | $0.53 \pm 0.78$ | $1.60 \pm 0.08$ | $0.36 \pm 0.62$ | $1.04 \pm 0.15$ |

Table 1: Distance to the goal for the point mass navigation task where numbers are mean and standard error for 3 seeds and 100 evaluation episodes per seed. Train is policy trained during reward learning. MaxEnt does not learn a policy during reward learning thus its performance is "NA". Eval (Train) uses the learned reward to train a policy from scratch on the same distribution used to train the reward. Eval (Test) measures the ability of the learned reward to generalize to a new starting state distribution.

Next, we report quantitative results in Table 1. We evaluate the performance of the policy trained at test time by reporting the distance from the policy's final trajectory state $s_T$ to the goal $g$: $\|s_T - g\|_2^2$. We report the final train performance of the algorithm ("Train"), along with the performance of the policy trained from scratch with the learned reward in the train distribution "Eval (Train)" and testing distribution "Eval (Test)". These results confirm that BC-IRL learns more generalizable rewards than baselines. Specifically, BC-IRL achieves a lower distance on the testing starting state distribution at 0.04, compared to 0.53, 1.6, and 0.36 for AIRL, GCL, and MaxEnt respectively. Surprisingly, BC-IRL even performs better than exact MaxEnt, which uses privileged information about the state space to estimate the partition function. This fits with our hypothesis that our method learns more generalizable rewards than MaxEnt, even when the MaxEnt objective is exactly computed. We repeat this analysis for a version of the task with an obstacle blocking the path to the goal in Appendix A.2 and reach the same findings even when BC-IRL must learn an asymmetric reward function. We also compare learned rewards to manually defined rewards in Appendix A.3.

Despite baselines learning rewards that do not generalize beyond the demonstrations, with enough environment interactions, policies trained under these rewards will eventually reach the high-rewards along the expert demonstrations. Since all demonstrations reach the goal in the point mass task, the X-shaped reward baselines learn have high-reward at the center. Despite the X-shaped providing little reward shaping off the X, with enough environment interactions, the agent eventually discovers the high-reward point at the goal. After training AIRL for 15M steps, 3x the number of steps for reward learning and the experiments in Table 1 and Figure 2, the policy eventually reaches $0.08 \pm 0.01$ distance to the goal. In the same setting, BC-IRL achieves $0.04 \pm 0.01$ distance to the goal in under 5M steps. The additional performance gap is due to BC-IRL learning a reward with a maximum reward value closer to the center (0.02 to the center) compared to AIRL (0.04 to the center).

## 5 EXPERIMENTS

In our experiments, we aim to answer the following questions: (1) Can BC-IRL learn reward functions that can train policies from scratch? (2) Does BC-IRL learn rewards that can generalize to unseen states and goals better than IRL baselines in complex environments? (3) Can learned rewards transfer better than policies learned directly with imitation learning? We show the first in Section 5.1 and the next two in Section 5.2. We evaluate on two continuous control tasks: 1) Fetch reaching task Szot et al. (2021) (Fig 3a), and the TriFinger reaching task Ahmed et al. (2021) (Fig 3b).

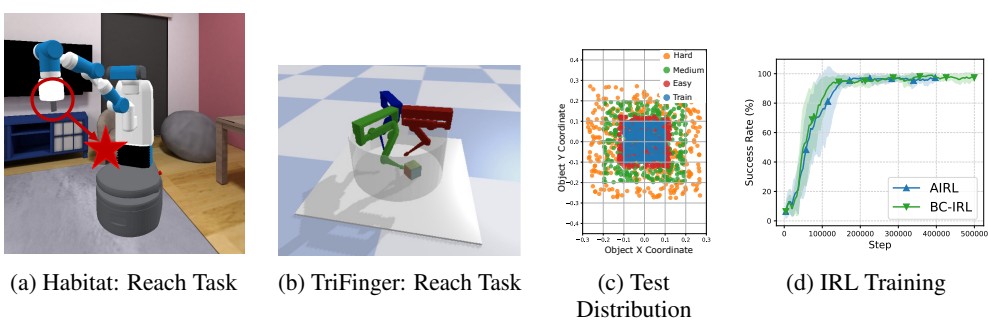

| (a) Habitat: Reach Task | (b) TriFinger: Reach Task | (c) Test Distribution | (d) IRL Training |
|---|---|---|---|

Figure 3: (a+b) Visualization of the Fetch and TriFinger reach tasks. c) 2D cross-section of the end-effector goal sampling regions in the reaching task. The reward function is trained on goals from the blue points; the learned reward must train policies to accomplish goals from Easy, Medium, and Hard test distributions of orange, green, and red points. d) Training curves during reward learning on Habitat Task, all methods succeed in training.

## 5.1 REWARD TRAINING PHASE: LEARNING REWARDS TO MATCH THE EXPERT

**Experimental Setup and Evaluation Metrics**   In the Fetch reaching task, setup in the Habitat 2.0 simulator Szot et al. (2021), the robot must move its end-effector to a 3D goal location $g$ which changes between episodes. The action space of the agent is the desired velocities for each of the 7 joints on the robot arm. The robot succeeds if the end-effector is within 0.1m of the target position by the 20 time step maximum episode length. During reward learning, the goal $g$ is sampled from a 0.2 meter length unit cube in front of the robot, $g \sim \mathcal{U}([0]^3, [0.2]^3)$. We provide 100 demonstrations.

|  | BC-IRL-PPO | AIRL |
|---|---|---|
| **Fetch Reach (Success)** $\uparrow$ | $1.00 \pm 0.00$ | $0.96 \pm 0.00$ |
| **Trifinger Reach (Goal Dist)** $\downarrow$ | $0.002 \pm 0.0015$ | $0.007 \pm 0.0017$ |

Table 2: Success rates for Fetch Reach and distance to goal for Trifinger Reach tasks in training policies to achieve the goal in the same start state and goal distributions as the expert demonstrations. Averages and standard deviations are from 3 seeds on Fetch Reach, and 5 seeds on Trifinger Reach with 100 episodes per seed.

For the Trifinger reaching task, each finger must move its fingertip to a 3D goal position. The fingers must travel a different distance and avoid getting blocked by another finger. Each finger has 3 joints, creating a 9D action and state space. The fingers are joint position controlled. We use a time horizon of $T = 5$ time steps. We provide a single demonstration. We report the final distance to the demonstrated goal, $(g - g^{\text{demo}})^2$ in meters.

**Evaluation and Baselines**   We evaluate BC-IRL-PPO by how well the reward it can train new policies from scratch in the same start state and goal distribution as the demonstrations. Given the pointmass results Section 4, we compare BC-IRL-PPO to AIRL, the best performing baseline for reward learning. More details on baseline choice, policy and reward representation, and hyperparameters are described in the Appendix (D).

**Results and Analysis**   As Table 2 confirms, our method and baselines are able to imitate the demonstrations when policies are evaluated in the same task setting as the expert. All methods are able to achieve a near 100% success rate and low distance to goal. Methods also learn with similar sample efficiency as shown in the learning curves in Figure 3d. These high-success rates indicate BC-IRL-PPO and AIRL learn rewards that capture the expert behavior and train policies to mimic the expert. When training policies in the same state/goal distribution as the expert, rewards from BC-IRL-PPO follow any constraints followed by the experts, just like the IRL baselines.

## 5.2 TEST PHASE: EVALUATING REWARD AND POLICY GENERALIZATION

In this section, we evaluate how learned rewards and policies can generalize to new task settings with increased starting state and goal sampling noise. We evaluate the generalization ability of rewards by evaluating how well they can train new policies to reach the goal in new start and goal distributions not seen in the demonstrations. This evaluation captures the reality that it is infeasible to collect demonstrations for every possible start/goal configuration. We thus aim to learn rewards from demonstrations that can generalize beyond the start/goal configurations present in those demonstrations. We quantify reward generalization ability by whether the reward can train a policy to perform the task in the new start/goal configurations.

For the Fetch Reach task, we evaluate on three wider test goal sampling distributions $g \sim \mathcal{U}([0]^3, [g_{\text{max}}]^3)$: Easy ($g_{\text{max}} = 0.25$), Medium ($g_{\text{max}} = 0.4$), and Hard ($g_{\text{max}} = 0.55$), all visualized in Figure 3c. Similarly, we evaluate on new state regions, which increase the starting and goal initial state distributions but exclude the regions from training, exposing the reward to only unseen initial states and goals. In Trifinger, we sample start configurations from around the start joint position in the demonstrations, with increasingly wider distributions ($s_0 \sim \mathcal{N}(s_0^{\text{demo}}, \delta)$, with $\delta = 0.01, 0.03, 0.05$).

We evaluate reward function performance by how well the reward function can train new policies from scratch. However, now the reward must generalize to inferring rewards in the new start state and goal distributions. We additionally compare to two imitation learning baselines: Generative Adversarial Imitation Learning (GAIL) Ho & Ermon (2016) and Behavior Cloning (BC). We compare different methods of transferring the learned reward and policy to the test setting:

**1) Reward**: Transfer only the reward from the above training phase and train a newly initialized policy in the test setting.

| | BC-IRL-PPO (Reward) | AIRL (Reward) | BC-IRL-PPO (Policy) | AIRL (Policy) | BC (Policy) | GAIL (Policy) |
|---|---|---|---|---|---|---|
| **Start Distrib: Easy** ($g_{\mathbf{max}} = 0.25$) | $\mathbf{1.00 \pm 0.00}$ | $0.96 \pm 0.04$ | $0.94 \pm 0.03$ | $0.89 \pm 0.01$ | $0.86 \pm 0.00$ | $0.88 \pm 0.02$ |
| **Start Distrib: Medium** ($g_{\mathbf{max}} = 0.4$) | $\mathbf{1.00 \pm 0.00}$ | $0.94 \pm 0.03$ | $0.70 \pm 0.10$ | $0.57 \pm 0.03$ | $0.56 \pm 0.01$ | $0.58 \pm 0.02$ |
| **Start Distrib: Hard** ($g_{\mathbf{max}} = 0.55$) | $\mathbf{0.76 \pm 0.16}$ | $0.34 \pm 0.01$ | $0.48 \pm 0.06$ | $0.38 \pm 0.04$ | $0.35 \pm 0.01$ | $0.37 \pm 0.04$ |
| **State Region: Easy** ($g_{\mathbf{max}} = 0.25$) | $\mathbf{1.00 \pm 0.00}$ | $0.98 \pm 0.04$ | $0.87 \pm 0.09$ | $0.74 \pm 0.02$ | $0.73 \pm 0.00$ | $0.76 \pm 0.04$ |
| **State Region: Medium** ($g_{\mathbf{max}} = 0.4$) | $\mathbf{1.00 \pm 0.00}$ | $0.88 \pm 0.10$ | $0.70 \pm 0.06$ | $0.58 \pm 0.03$ | $0.52 \pm 0.01$ | $0.54 \pm 0.01$ |
| **State Region: Hard** ($g_{\mathbf{max}} = 0.55$) | $\mathbf{0.78 \pm 0.13}$ | $0.34 \pm 0.04$ | $0.49 \pm 0.08$ | $0.42 \pm 0.02$ | $0.39 \pm 0.02$ | $0.42 \pm 0.03$ |

Table 3: Success rates for the reaching task comparing the generalization capabilities of IRL and imitation learning methods. "(Reward)" transfers the learned reward from IRL methods and trains a newly initialized policy in the test setting. "(Policy)" transfers the policy without training in the new setting. The Easy, Medium, and Hard indicate the difficulty of generalization where the end-effector goal is sampled from $g \sim \mathcal{U}([0]^3, [g_{\max}]^3)$.

**2) Policy**: Transfer only the policy from the above training phase and immediately evaluate the policy without further training in the test setting. This compares transferring learned rewards and transferring learned policies. We use this transfer strategy to compare against direct imitation learning methods.

**3) Reward+Policy**: Transfer the reward and policy and then fine-tune the policy using the learned reward in the test setting. Results for this setting are in Appendix B.2.

**Results and Analysis**    The results in Table 3 show BC-IRL-PPO learns rewards that generalize better than IRL baselines to new settings. In the hardest generalization setting, BC-IRL-PPO achieves over twice the success rate of AIRL. AIRL struggles to transfer its learned reward to harder generalization settings, with performance decreasing as the goal sampling distribution becomes larger and has less overlap with the training goal distribution. In the "Hard" start region generalization setting, the performance of AIRL degrades to 34% success rate. On the other hand, BC-IRL-PPO learns a generalizable reward and performs well even in the "Hard" generalization strategy, achieving 76% success. This trend is true both for generalization to new start state distributions and for new start state regions. The results for Trifinger Reach in Table 4 support these findings with rewards learned via BC-IRL-PPO generalizing better to training policies from scratch in all three test distributions. All training curves for training policies from scratch with learned rewards are in Appendix B.1.

Furthermore, the results in Table 3 also demonstrate that transferring rewards "(Reward)" is more effective for generalization than transferring policies "(Policy)". Transferring the reward to train new policies typically outperforms transferring only the policy for all IRL approaches. Additionally, training from scratch with rewards learned via IRL outperforms non-reward learning imitation learning methods that only permit transferring

| | BC-IRL-PPO | AIRL |
|---|---|---|
| **Test** $\delta = 0.01$ | $\mathbf{0.0065} \pm \mathbf{0.002}$ | $0.012 \pm 0.0017$ |
| **Test** $\delta = 0.03$ | $\mathbf{0.0061} \pm \mathbf{0.002}$ | $0.012 \pm 0.0008$ |
| **Test** $\delta = 0.05$ | $\mathbf{0.0061} \pm \mathbf{0.001}$ | $0.0117 \pm 0.0015$ |

Table 4: Distance to the goal for Trifinger reach, evaluating how rewards generalize to training policies in new start/ goal distributions.

the policy zero-shot. The policies learned by GAIL and BC generalize worse than training new policies from scratch with the reward learned by BC-IRL-PPO, with BC and GAIL achieving 35% and 37% success rates in the "Hard" generalization setting while our method achieves 76% success. The superior performance of BC-IRL-PPO over BC highlights the important differences between the two methods with our method learning a reward and training the policy with PPO on the learned reward.

In Appendix B.2, we also show the "Policy+Reward" transfer setting and demonstrate BC-IRL-PPO also outperforms baselines in this setting. In Appendix B we also analyze performance with the number of demos, different inner and outer loop learning rates, and number of inner loop updates.

## 6 DISCUSSION AND FUTURE WORK

We propose a new IRL framework for learning generalizable rewards with bi-level gradient-based optimization. By meta-learning rewards, our framework can optimize alternative outer-level objectives instead of the maximum entropy objective commonly used in prior work. We propose BC-IRL-PPO an instantiation of our new framework, which uses PPO for policy optimization in the inner loop and an action matching objective in the outer loop. We demonstrate that BC-IRL-PPO learns rewards that generalize better than baselines. Potential negative social impacts of this work are that learning reward functions from data could result in less interpretable rewards, leading to more opaque behaviors from agents that optimize the learned reward.

Future work will explore alternative instantiations of the BC-IRL framework, such as utilizing sample efficient off-policy methods like SAC or model-based methods in the inner loop. Model-based methods are especially appealing because a single dynamics model could be shared between tasks and learning reward functions for new tasks could be achieved purely using the model. Finally, other outer loop objectives rather than action matching are also possible.

## 7 ACKNOWLEDGMENTS

The Georgia Tech effort was supported in part by NSF, ONR YIP, and ARO PECASE. The views and conclusions contained herein are those of the authors and should not be interpreted as necessarily representing the official policies or endorsements, either expressed or implied, of the U.S. Government, or any sponsor.

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

## A    FURTHER POINT MASS NAVIGATION RESULTS

### A.1    QUALITATIVE RESULTS FOR ALL METHODS IN POINT MASS NAVIGATION

Visualizations of the reward functions from all methods for the regular pointmass task are displayed in Figure 4.

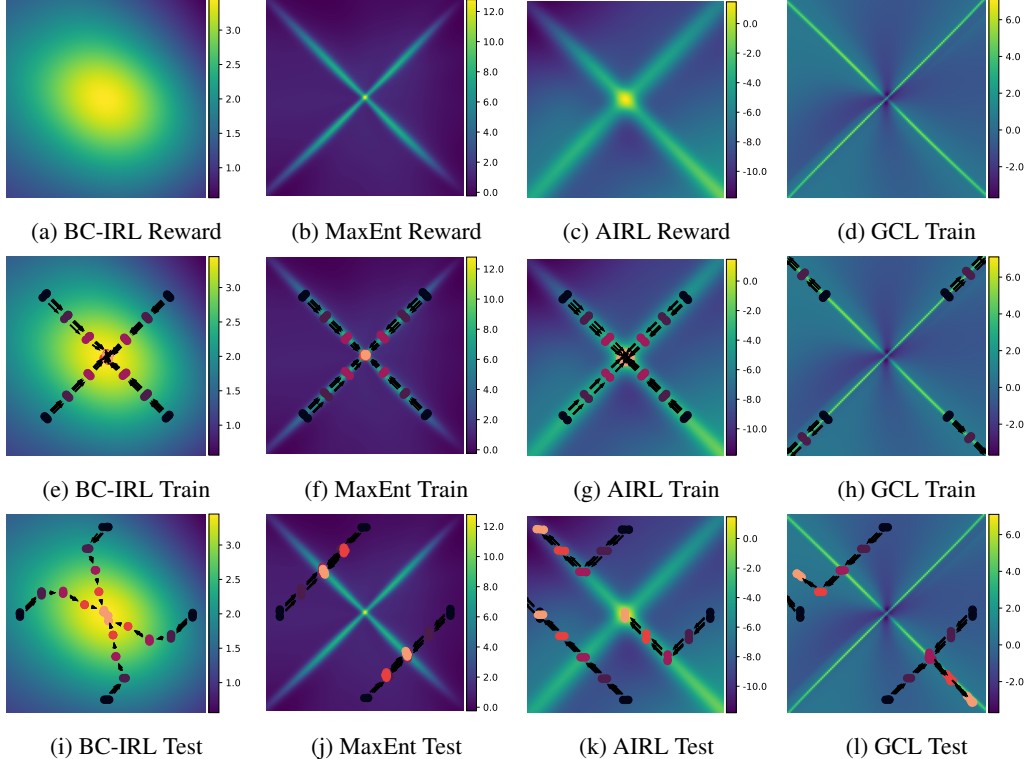

Figure 4: Qualitative results for all methods on the point mass navigation task without the obstacle.

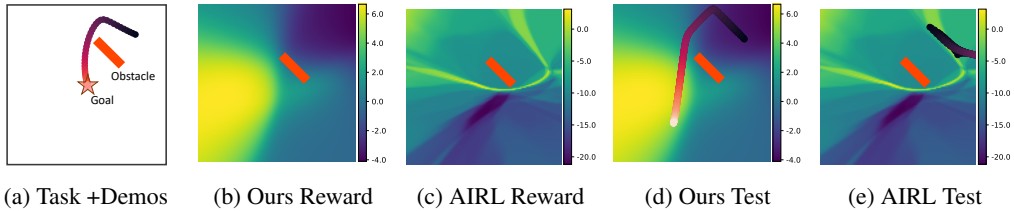

Figure 5: Results on the point mass navigation tasks (top row: no-obstacle, bottom row: obstacle task). We show the learned reward functions of our method (b, g) vs. AIRL (c, h) and the policies learned from scratch using those reward functions (d, e, i, j).

## A.2   OBSTACLE POINT MASS NAVIGATION

The obstacle point mass navigation task incorporates asymmetric dynamics with an off-centered obstacle. This environment is the same as the point mass navigation task from Section 4, except there is an obstacle blocking the path to the center and the agent only spawns in the top-right hand corner. This task has a trajectory length of $T = 50$ time steps with 100 demonstrations. Figure 5a visualizes the expert demonstrations where darker points are earlier time steps.

|  | BC-IRL | AIRL | GCL | MaxEnt | f-IRL |
|---|---|---|---|---|---|
| **Train** | $0.08 \pm 0.00$ | $0.62 \pm 0.50$ | $1.30 \pm 0.71$ | NA | $0.41 \pm 0.35$ |
| **Eval (Train)** | $0.62 \pm 0.60$ | $1.42 \pm 0.17$ | $2.02 \pm 0.18$ | $1.07 \pm 0.39$ | $0.61 \pm 0.33$ |
| **Eval (Test)** | $\mathbf{0.79} \pm \mathbf{0.65}$ | $1.42 \pm 0.18$ | $2.01 \pm 0.18$ | $0.83 \pm 0.01$ | $1.53 \pm 0.24$ |

Table 5: Distance to the goal for the point mass navigation task where numbers are mean and standard error for 3 seeds and 100 evaluation episodes per seed. Train is policy trained during reward learning. MaxEnt does not learn a policy during reward learning thus its performance is "NA". Eval (Train) uses the learned reward to train a policy from scratch on the same distribution used to train the reward. Eval (Test) measures the ability of the learned reward to generalize to a new starting state distribution.

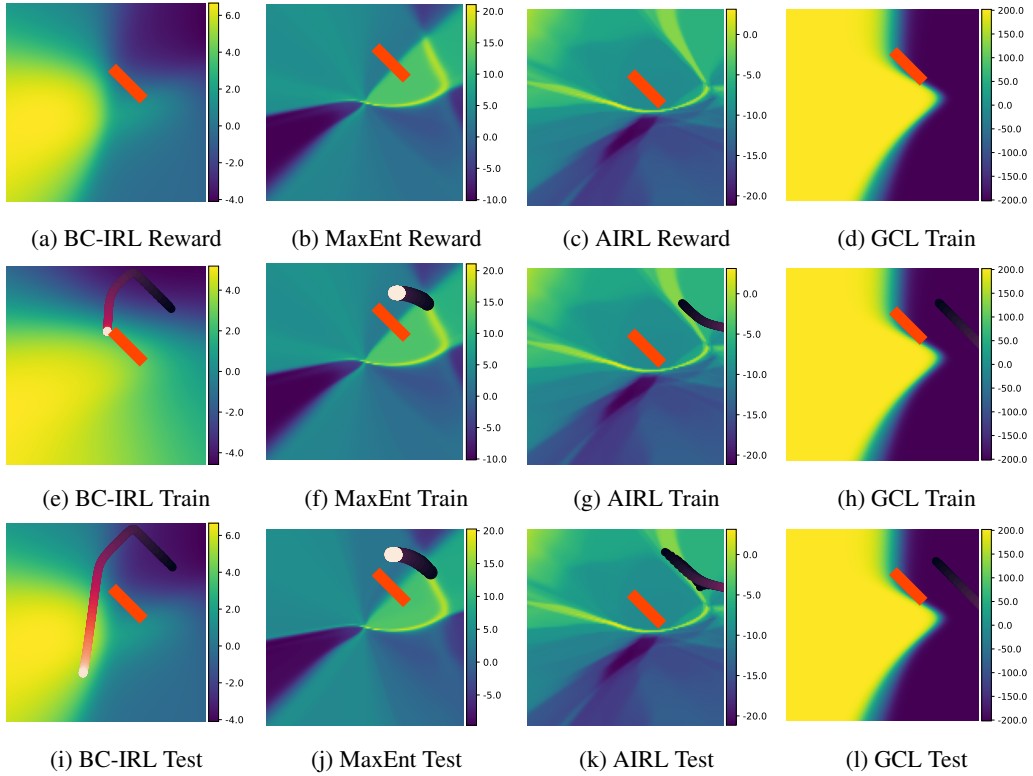

Figure 6: Qualitative results for all methods on the point mass navigation task with the obstacle.

The results in Table 5 are consistent with the non-obstacle point mass task where BC-IRL generalizes better than a variety of MaxEnt IRL baselines. In the train setting, BC-IRL learns rewards that match the expert behavior with avoiding the obstacle and even achieves better performance than baselines in this task with 0.08 distance to the goal versus 0.41 to the goal for the best performing baseline in the train setting, f-IRL. BC-IRL generalizes better than baselines achieving 0.79 distance to goal compared to the best performing baseline MaxEnt, which also has access to oracle information. The reward learned by BC-IRL visualized in Figure 5b shows BC-IRL learns a complex reward to account for the obstacle. Figure 6 visualizes the rewards for all methods.

### A.3 COMPARISON TO MANUALLY DEFINED REWARDS

We compare the rewards learned by BC-IRL to two hand-coded rewards. We visualize how well the learned rewards can train policies from scratch in the evaluation distribution in the point navigation with obstacle task. The reward learned by BC-IRL therefore must generalize. On the other hand, the hand-coded rewards do not require any learning. We include a sparse reward for achieving the goal, which does not require domain knowledge when implementing the reward. We also implement a dense reward, defined as the change in Euclidean distance to the goal where $r_t = d_{t-1} - d_t$ where $d_t$ is the distance of the agent to the goal at time $t$.

Figure 7a shows policy training curves for the learned and hand-defined rewards. The sparse reward performs poorly and the policy fails to get closer to the goal. On the other hand, the rewards learned by BC-IRL guide the policy closer to the goal. The dense reward, which incoporates more domain knowledge about the task, performs better than the learned reward.

### A.4 ANALYZING NUMBER OF INNER LOOP UPDATES

As described in Section 3.3, a hyperparameter in BC-IRL-PPO is the number of inner loop policy optimization steps $K$, for each reward function update. In our experiments, we selected $K = 1$. In Figure 7b we examine the training performance of BC-IRL-PPO in the point navigation task with no obstacle for various choices of $K$. We find that a wide variety of $K$ values perform similarly. We,

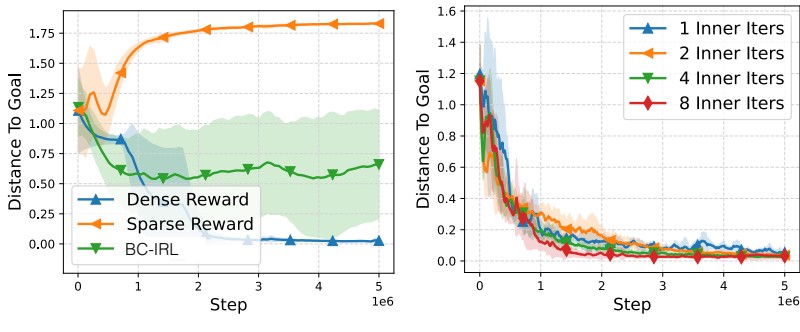

(a) Hand-coded rewards vs. BC-IRL.  (b) # inner loop updates in BC-IRL.

Figure 7: Left: Comparing the reward learned from BC-IRL to two manually hand-coded rewards. Right: Comparing different number of inner loop steps in BC-IRL.

therefore, selected $K = 1$ since it runs the fastest, with no need to track multiple policy updates in the meta optimization.

### A.5 BC-IRL with Model-Based Policy Optimization

We compare BC-IRL-PPO to a version of BC-IRL that uses model-based RL in the inner loop inspired by Das et al. (2020). A direct comparison to Das et al. (2020) is not possible because their method assumes access to a pre-trained dynamics model, while in our work, we do not assume access to a ground truth or pre-trained dynamics model. However, we compare to a version of Das et al. (2020) in the point mass navigation task with a ground truth dynamics model. Specifically, we use gradient-based MPC in the inner loop optimization as in Das et al. (2020), but the BC IRL outer loop objective. With the BC outer loop objective, it also learns generalizable rewards in the point mass navigation task achieving $0.06 \pm 0.03$ distance to goal in "Eval (Train)" and $0.07 \pm 0.03$ in "Eval (Test)". However, in the point mass navigation task with the obstacle, this method fails to learn a reward and struggles to minimize the outer loop objective. We hypothesize that in longer horizon tasks, the MPC inner loop optimization in [9] easily gets stuck in local minimas and struggles to differentiate through the entire MPC optimization.

## B Reach Task: Further Experiment Results

### B.1 RL-Training Curves

In Figure 8 we visualize the training curves for the RL training used in Table 3. Figure 8a shows policy learning progress during the IRL training phase. In each setting, the performance is measured by using the current reward to train a policy and computing the success rate of the policy. Figure 8b to Figure 8d show the policy learning curves at test time, in the generalization settings, where the reward is frozen and must generalize to learn new policies on new goals ("Reward " transfer strategy). These plots show that all methods learn similarly during IRL training (Figure 8a). When transferring the learned rewards to test settings we see that BC-IRL-PPO performs better in training successful policies as the generalization difficulty increases with the most difficult generalization in Figure 8d.

### B.2 Transfer Reward+Policy Setting

| | BC-IRL-PPO (Policy+Reward) | BC-IRL-PPO (Reward) | AIRL (Policy+Reward) | AIRL (Reward) |
|---|---|---|---|---|
| **Train** ($g_{\mathbf{max}} = 0.2$) | $1.00 \pm 0.00$ | $1.00 \pm 0.00$ | $0.96 \pm 0.00$ | $0.96 \pm 0.00$ |
| **Start Distrib: Easy** ($g_{\mathbf{max}} = 0.25$) | $\mathbf{1.00} \pm \mathbf{0.00}$ | $\mathbf{1.00} \pm \mathbf{0.00}$ | $0.92 \pm 0.00$ | $0.96 \pm 0.04$ |
| **Start Distrib: Medium** ($g_{\mathbf{max}} = 0.4$) | $\mathbf{1.00} \pm \mathbf{0.00}$ | $\mathbf{1.00} \pm \mathbf{0.00}$ | $0.81 \pm 0.07$ | $0.93 \pm 0.05$ |
| **Start Distrib: Hard** ($g_{\mathbf{max}} = 0.55$) | $\mathbf{0.80} \pm \mathbf{0.14}$ | $0.76 \pm 0.16$ | $0.38 \pm 0.03$ | $0.38 \pm 0.06$ |
| **State Region: Easy** ($g_{\mathbf{max}} = 0.25$) | $\mathbf{1.00} \pm \mathbf{0.00}$ | $\mathbf{1.00} \pm \mathbf{0.00}$ | $0.05 \pm 0.02$ | $\mathbf{1.00} \pm \mathbf{0.01}$ |
| **State Region: Medium** ($g_{\mathbf{max}} = 0.4$) | $\mathbf{1.00} \pm \mathbf{0.00}$ | $\mathbf{1.00} \pm \mathbf{0.00}$ | $0.17 \pm 0.09$ | $0.91 \pm 0.02$ |
| **State Region: Hard** ($g_{\mathbf{max}} = 0.55$) | $0.76 \pm 0.15$ | $\mathbf{0.78} \pm \mathbf{0.13}$ | $0.21 \pm 0.11$ | $0.32 \pm 0.08$ |

Table 6: Results for the "Policy+Reward" transfer strategy where the trained policy and reward are transferred to the test setting and the policy is fine-tuned.

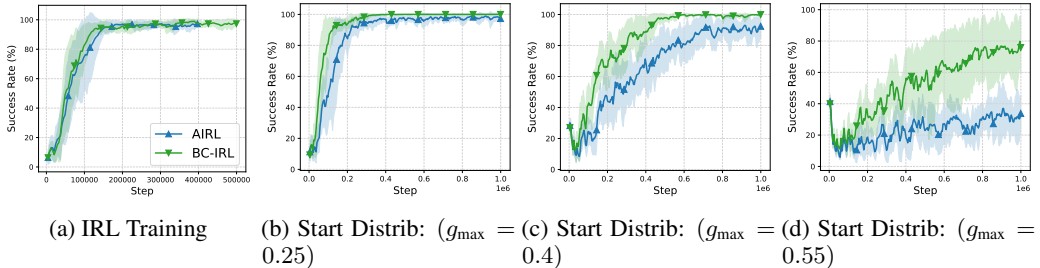

(a) IRL Training    (b) Start Distrib: ($g_{max} =$ (c) Start Distrib: ($g_{max} =$ (d) Start Distrib: ($g_{max} =$
0.25)                    0.4)                     0.55)

Figure 8: Learning curves for the training setting and "Reward" transfer strategies from Table 3. All results are for 3 seeds and error regions show the standard deviation in success rate between seeds.

Here, we evaluate the **"Policy+Reward"** transfer strategy to new environment settings where both the reward and policy are transferred. In the new setting, "Policy+Reward" uses the transferred reward to fine-tune the pre-trained transferred policy with RL. We show results in Table 6 for the "Policy+Reward" transfer strategy alongside the "Reward" transfer strategy from Table 3. We find that "Policy+Reward" performs slightly better than "Reward" in the Hard setting of generalization to new starting state distributions but otherwise performs similarly. Even in the "Policy+Reward" setting, AIRL struggles to learn a good policy in the Medium and Hard settings, achieving 38% and 81% success rate respectively.

### B.3 ANALYZING THE NUMBER DEMONSTRATIONS

|  | BC-IRL-PPO (Train) | BC-IRL-PPO ($g_{max} = 0.4$) | BC-IRL-PPO ($g_{max} = 0.55$) | AIRL (Train) | AIRL ($g_{max} = 0.4$) | AIRL ($g_{max} = 0.55$) |
|---|---|---|---|---|---|---|
| **100 Demos** | $1.00 \pm 0.0$ | $1.00 \pm 0.00$ | $0.76 \pm 0.16$ | $0.96 \pm 0.00$ | $0.93 \pm 0.05$ | $0.38 \pm 0.06$ |
| **50 Demos** | $0.96 \pm 0.05$ | $0.91 \pm 0.16$ | $0.70 \pm 0.17$ | $0.96 \pm 0.01$ | $0.88 \pm 0.09$ | $0.40 \pm 0.05$ |
| **25 Demos** | $0.99 \pm 0.01$ | $0.99 \pm 0.01$ | $0.56 \pm 0.12$ | $0.95 \pm 0.01$ | $0.84 \pm 0.03$ | $0.48 \pm 0.09$ |
| **5 Demos** | $0.99 \pm 0.01$ | $1.00 \pm 0.01$ | $0.69 \pm 0.19$ | $0.97 \pm 0.01$ | $0.84 \pm 0.03$ | $0.42 \pm 0.02$ |

Table 7: Comparing the number of demonstrations for BC-IRL-PPO and AIRL across the train, medium, and hard settings.

We analyze the effect of the number of demonstrations used for reward learning in Table 7. We find that using fewer demonstrations does not affect the training performance of BC-IRL-PPO and AIRL. We also find our method does just as well with 5 demos as 100 in the +75% noise setting, with any number of demonstrations achieving near-perfect success rates. On the other hand, the performance of AIRL degrades from 93% success rate with 100 demonstrations to 84% in the +75% noise setting. In the +100% noise setting, fewer demonstrations hurt performance for both methods, with our method dropping from 76% success to 69% success and AIRL from 38% success to 42% success.

### B.4 BC-IRL HYPERPARARAMETER ANALYSIS

BC-IRL-PPO requires a learning rate for the policy optimization and a learning rate for the reward optimization. We compare the performance of our algorithm for various choices of policy and reward learning rates in Table 8. We find that across many different learning rate settings our method achieves high rates of success, but high policy learning rates have a detrimental effect. High reward learning rates have a slight negative impact but are not as severe.

## C FURTHER 2D POINT NAVIGATION DETAILS

The start state distributions for the 2D point navigation task are illustrated in Figure 9. The reward is learned using the start distribution in red on 4 equally spaced points from the center. Four demonstrations are also provided in this train start state distribution from each of the four corners. The reward is then transferred and a new policy is trained with the start state distribution in the magenta color. This start state distribution has no overlap with the train distribution and is also equally spaced. The reward must therefore generalize to providing rewards in this new state distribution.

The hyperparameters for the methods from the 2D point navigation task in Section 4 are detailed in Table 9 for the no obstacle version and Table 10 for the obstacle version of the task. The reward

|  | Reward LR | | | |
| Policy LR | 1e-4 | 1e-3 | 1e-2 | 1e-1 |
|---|---|---|---|---|
| 1e-4 | $0.96 \pm 0.01$ | $0.96 \pm 0.01$ | $0.89 \pm 0.11$ | $0.71 \pm 0.45$ |
| 1e-3 | $0.97 \pm 0.03$ | $1.00 \pm 0.00$ | $1.00 \pm 0.00$ | $0.66 \pm 0.56$ |
| 1e-2 | $0.14 \pm 0.23$ | $0.28 \pm 0.48$ | $0.11 \pm 0.17$ | $0.59 \pm 0.53$ |
| 1e-1 | $0.64 \pm 0.55$ | $0.33 \pm 0.55$ | $0.02 \pm 0.04$ | $0.32 \pm 0.54$ |

Table 8: Comparing choice of learning rate for the inner and outer loops for BC-IRL-PPO on the train setting. Numbers display success rate.

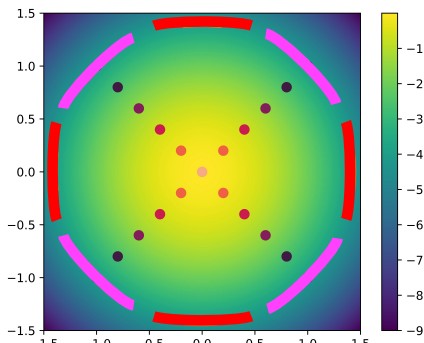

Figure 9: The starting state distribution for the 2D point navigation task with the demonstrations and negative distance to the goal overlaid. The training start state distribution where the reward is learned is in red. The test start state distribution where the reward is transferred is in magenta.

function / discriminator for all methods was a neural network with 1 hidden layer and 128 hidden dimension size with $tanh$-activations between the layers. Adam Kingma & Ba (2014) was used for policy and reward optimization. All RL training used 5M steps of experience for the training and testing setting for the navigation no obstacle task. f-IRL uses the same optimization and neural network hyperparameters for the discriminator and reward function. Like in Ni et al. (2020), we clamp the output of the reward function within the range $[-10, 10]$ and found this was beneficial for learning. In the navigation with obstacle task, training used 15M steps of experience and testing used 5M steps of experience. All experiments were run on a Intel(R) Core(TM) i9-9900X CPU @ 3.50GHz.

| Hyperparameter | BC-IRL-PPO | AIRL | GCL | MaxEnt | f-IRL |
|---|---|---|---|---|---|
| Reward Learning Rate | 1e-4 | 1e-3 | 3e-4 | 1e-3 | 3e-4 |
| Reward Batch Size | 20 | 20 | 20 | 20 | 20 |
| Policy Learning Rate | 1e-4 | 1e-4 | 3e-4 | 3e-4 | 3e-4 |
| Policy Learning Rate Decay | True | True | True | False | False |
| Policy # Mini-batches | 4 | 4 | 4 | 4 | 4 |
| Policy # Epochs per Update | 4 | 4 | 4 | 4 | 4 |
| Policy Entropy Coefficient | 1e-4 | 1e-4 | 1e-4 | 1e-4 | 1e-4 |
| Discount Factor $\gamma$ | 0.99 | 0.99 | 0.99 | 0.99 | 0.99 |
| Policy Batch Size | 1280 | 1280 | 1280 | 1280 | 1280 |

Table 9: 2D navigation without obstacle method hyperparameters. These hyperparameters were used both in the training and reward transfer settings.

| Hyperparameter | BC-IRL-PPO | AIRL | GCL | MaxEnt |
|---|---|---|---|---|
| Reward Learning Rate | 1e-4 | 1e-3 | 3e-4 | 1e-3 |
| Reward Batch Size | 256 | 256 | 256 | 256 |
| Policy Learning Rate | 3e-4 | 3e-4 | 3e-4 | 3e-4 |
| Policy Learning Rate Decay | True | True | True | False |
| Policy # Mini-batches | 4 | 4 | 4 | 4 |
| Policy # Epochs per Update | 4 | 4 | 4 | 4 |
| Policy Entropy Coefficient | 1e-4 | 1e-4 | 1e-4 | 1e-4 |
| Discount Factor $\gamma$ | 0.99 | 0.99 | 0.99 | 0.99 |
| Policy Batch Size | 6400 | 6400 | 6400 | 6400 |

Table 10: 2D navigation with obstacle method hyperparameters. These hyperparameters were used both in the training and reward transfer settings.

| Hyperparameter | BC-IRL-PPO | AIRL | GAIL | BC |
|---|---|---|---|---|
| Reward Learning Rate | 3e-4 | 1e-4 | 1e-4 | NA |
| Reward Batch Size | 128 | 128 | 128 | NA |
| Policy Learning Rate | 3e-4 | 1e-4 | 1e-4 | 1e-4 |
| Policy Learning Rate Decay | False | True | True | False |
| Policy # Mini-batches | 4 | 4 | 4 | NA |
| Policy # Epochs per Update | 4 | 4 | 4 | NA |
| Policy Entropy Coefficient | 0.0 | 0.0 | 0.0 | NA |
| Discount Factor $\gamma$ | 0.99 | 0.99 | 0.99 | 0.99 |
| Policy Batch Size | 4096 | 4096 | 4096 | NA |

Table 11: Method hyperparameters for the Fetch reaching task. These hyperparameters were used both in the training and reward transfer settings.

## D  FURTHER REACH TASK DETAILS

### D.1  CHOICE OF BASELINES

The "Exact MaxEntIRL" approach is excluded because it cannot be computed exactly for high-dimensional state spaces. GCL is excluded because of its poor performance on the toy task relative to other methods. We also compare to the following imitation learning methods which learn only policies and no transferable reward:

- **Behavioral Cloning (BC)** Bain & Sammut (1995): Train a policy using supervised learning to match the actions in the expert dataset.
- **Generative Adversarial Imitation Learning (GAIL)** Ho & Ermon (2016): Trains a discriminator to distinguish expert from agent transitions and then use the discriminator confusion score as the reward. This reward is coupled with the current policy Finn et al. (2016a) (referred to as a "pseudo-reward") and therefore cannot train policies from scratch.

### D.2  POLICY+NETWORK REPRESENTATION

All methods use a neural network to represent the policy and reward with 1 hidden layer, 128 hidden units, and $tanh$-activation functions between the layers. We use PPO as the policy optimization method for all methods. All methods in all tasks use demonstrations obtained from a policy trained with PPO using a manually engineered reward.

### D.3  HYPERPARAMETERS

The hyperparameters for all methods from the Reaching task are described in Table 11. The Adam optimizer Kingma & Ba (2014) was used for policy and reward optimization. All RL training used 1M steps of experience for the training and testing settings. The "Reward" and "Policy+Reward" transfer strategies trained policies with the same set of hyperparameters.

| Hyperparameter | BC-IRL-PPO | AIRL |
|---|---|---|
| Reward Learning Rate | 1e-3 | 1e-3 |
| Reward Batch Size | 6 | 6 |
| Policy Learning Rate | 1e-4 | 1e-3 |
| Policy Learning Rate Decay | False | False |
| Policy # Mini-batches | 4 | 4 |
| Policy # Epochs per Update | 2 | 2 |
| Policy Entropy Coefficient | 0.005 | 0.005 |
| Discount Factor $\gamma$ | 0.99 | 0.99 |
| Policy Batch Size | 40 | 40 |

Table 12: Method hyperparameters for the Trifinger reaching task. These hyperparameters were used both in the training and reward transfer settings.

## E   TRIFINGER EXPERIMENT DETAILS

### E.1   POLICY+NETWORK REPRESENTATION

All methods use a neural network to represent the policy and reward with 1 hidden layer, 128 hidden units, and $tanh$-activation functions between the layers. We use PPO as the policy optimization method for all methods. All methods in all tasks use demonstrations obtained from a policy trained with PPO using a manually engineered reward.

### E.2   HYPERPARAMETERS

The hyperparameters for all methods for the Trifinger reaching task are described in Table 12. The Adam optimizer Kingma & Ba (2014) was used for policy and reward optimization. All RL training used 500k steps of experience for the reward training phase and 100k steps of experience for policy optimization in test settings.

