# OpenReview forum: "BC-IRL: Learning Generalizable Reward Functions from Demonstrations"
_ICLR.cc/2023/Conference — ICLR 2023 notable top 25%_

### Official Review · Reviewer_pza4 · 2022-10-24

**Confidence:** 4
**Correctness:** 2
**Technical Novelty And Significance:** 2
**Empirical Novelty And Significance:** 2
**Recommendation:** 3

**Clarity, Quality, Novelty And Reproducibility:**

Clarity and quality need to be greatly improved, see above for more detailed discussion of weaknesses.

**Strength And Weaknesses:**

Strengths:
- The paper proposes a novel algorithm for learning rewards with bi-level gradient-based optimization.

Weaknesses:
- Unclear in what sense the authors are referring to when they use the term “generalization”. In the experiments, the generalization seems like it is limited to slightly different start and goal distributions. This should be made clearer.
- GAIL is a method that trains a policy that matches the state-action distribution of the expert data, it should be clarified how the goal here is different since this method also aims to learn a policy that matches the behavior of the expert.
- Experimental evaluation is very limited. Only 2 tasks of a similar nature are evaluated, and the tasks are reaching, which is very simple. Also, the environments have a low-dimensional state space, despite the simplicity of the tasks.
- How does this compare to AIRL or GAIL where the discriminator learned uses mixup regularization or spectral norm? These are 2 common techniques (among others) for making the discriminator less brittle. Much more experimentation is needed to conclude how much more generalizable the rewards learned by BC-IRL are.


**Summary Of The Paper:**

Providing informative rewards is crucial for effective reinforcement learning. Prior IRL methods overfit to demonstrations and fail to learn generalizable rewards. To combat this issue, this paper proposes BC-IRL, which uses gradient-based bi-level optimization to learn the reward. The authors evaluate on two continuous control tasks against IRL and imitation learning methods.

**Summary Of The Review:**

Due to issues with clarity and quality, this paper does not seem ready for acceptance.

---

> ### Author Response · Authors · 2022-11-12
> **Response to Reviewer pza4**
>
> **1. Unclear in what sense the authors are referring to when they use the term “generalization”.**
>
> In the paper, we clarify the exact generalization conditions for each task.
>
> * For the point mass experiments, we state, “The testing starting state distribution has no overlap with the training start state distribution” (Sec. 4) and visualize the starting state distributions in Fig. 9.
> * We clarify in the paper for the Fetch experiments that “during reward learning, the goal $g$ is sampled from a 0.2 meter length unit cube in front of the robot” (Sec. 5.1) and “we evaluate on three wider test goal sampling distributions $g \sim \mathcal{U}([0]^3, [g_{\text{max}}]^3)$: Easy ($g_{\text{max}} = 0.25$), Medium ($g_{\text{max}} = 0.4$), and Hard ($g_{\text{max}} = 0.55$)” (Sec 5.2). Additionally, we visualize these distributions in Fig. 3c.
> * Finally, for the Trifinger task, we clarify the generalization settings as “we sample start configurations from around the start joint position in the demonstrations, with increasingly wider distributions ($s_0 \sim \mathcal{N}(s_0^\text{demo}, \delta)$, with $\delta =0.01, 0.03, 0.05$)”.
>
> A larger sampling noise value indicates an evaluation start state distribution more different from the train start state distribution. As the start state distribution changes, generalization becomes harder, and BC-IRL performs better than baselines.
>
> **2. How does this compare to AIRL or GAIL where the discriminator learned uses mixup regularization or spectral norm?**
>
> Thank you for the suggestion. We included mixup regularization for the reward function in the AIRL baseline. We ran experiments for 3 seeds in the point mass task from Fig. 2 with the same setting of using 5M steps for reward and policy learning. We find that on the “Eval (Test)” distribution, AIRL with mixup regularization achieves $0.46$ distance to the goal and AIRL without mixup regularization achieves $0.53$ distance to the goal. BC-IRL performs better than both with $0.04$ distance to the goal. While regularization can help prevent the reward from overfitting to the demonstrations, it cannot overcome the fundamental overfitting issues of MaxEnt IRL, and BC-IRL still generalizes better.
>
> **3. GAIL is a method that trains a policy that matches the state-action distribution of the expert data, it should be clarified how the goal here is different since this method also aims to learn a policy that matches the behavior of the expert.**
>
> The goal of BC-IRL is to learn generalizable reward functions. The goal of GAIL is to learn a policy, not a reward, since it is an imitation learning algorithm, not an inverse-RL algorithm. AIRL, an extension of GAIL for IRL, learns rewards. However, as discussed in Sec. 3.1, since AIRL is based on MaxEnt IRL, it learns rewards that overfit to the expert demonstrations. Through a fundamentally different objective, we show BC-IRL learns rewards that train policies twice as well in new generalization settings as baselines.

---

> ### Author Response · Authors · 2022-11-17
> **Has Our Response Addressed Your Concerns?**
>
> Dear Reviewer pza4, we would be grateful if you can comment on whether our response addressed your concerns or if issues remain.

---

### Official Review · Reviewer_9FVR · 2022-10-24

**Confidence:** 2
**Correctness:** 1
**Technical Novelty And Significance:** 4
**Empirical Novelty And Significance:** 3
**Recommendation:** 8

**Clarity, Quality, Novelty And Reproducibility:**

The paper is well written, the method is simple to understand and well described.
I have never seen such an approach for IRL before.


**Strength And Weaknesses:**

The method is novel, easy to understand and implement and the results looks impressive.

I however have a concern regarding the main tool-experiment as shown in Figure 1 and 2.
Even if the reward is only on the states that are occupied by the demonstration (resulting in a X-shape), the learned behaviour with such a reward should still learn to reach the highest rewarding states.

For example, a Value-function associated with such a reward would diffuse values outside of the X cross.
In this experiment there is no value-function, but they use PPO that directs policies updates with generalized advantage estimators which behave exactly like a value-function (it is a non-biased estimator). Therefor the fact there is no reward outside the cross should not significantly affect the optimal behaviour.

I would understand that the reward obtained by BC-IRL is much more dense, and so accelerates the learning. But AIRL well implemented should not fail on this task (at least with enough learning steps). Does AIRL ends up learning the good behaviour after a (much) longer training?



**Summary Of The Paper:**

This paper introduces a new form of IRL in which the learned reward is not based on the state occupancy, but is instead computed in order to lead a policy-gradient learner at imitating the demonstration.
This is done with a meta-learning approach, in which the inner loop updates the policy (with a policy-gradient objective) and the outer loop updates the reward (with a BC objective).
They observe that such a reward that is not based on occupancy is better at generalising behaviours and is more robust to changes of the initial distribution.


**Summary Of The Review:**

I think this paper is definitely novel and shows impressive improvements of SOTA approaches, but I have a strong concern regarding the implementations and behaviour of the baselines. In that doubt I'm giving a low score, but I am ready to significantly increase it if authors adresse my comments with a convincing explanation.


After reading authors answer and revisions, I've increased my socre.

---

> ### Author Response · Authors · 2022-11-12
> **Response to Reviewer 9FVR**
>
> **1. Even if the reward is X-shaped, the learned behavior with such a reward should still learn to reach the highest rewarding states**
>
> In the training distribution, baselines are able to reach the high reward states (rows 1,2 of Table 1). However, in the test distribution in Fig. 2, baselines fail to learn with the X-shaped reward. The X-shaped reward provides little reward shaping outside the X-shaped high reward region. At test time, all methods train policies with the learned reward using the same number of samples (5M, which is also the number of samples used in reward learning).
>
> For this rebuttal, we ran experiments to confirm that given more than 5M training steps, policy optimization will eventually converge to the high-reward area for the X-shaped rewards AIRL learns. We continued training for all 3 seeds on the point mass task from Fig. 2 on the “Eval (Test)” starting state distribution (row 3 of Table 1). After 15M steps of interactions, the distance to the goal in the “Eval (Test)” distribution eventually reaches $ 0.08 $ for AIRL. In the same setting, BC-IRL achieves $ 0.04 $ distance to the goal in under 5M steps. The additional performance gap is due to BC-IRL learning a reward with a maximum reward value closer to the center ($0.02$ to the center) compared to AIRL ($0.04$ to the center).
>
> **2. I have a strong concern regarding the implementations and behavior of the baselines**
>
> The strong performance of the baselines on the train distribution and easier generalization distributions verifies the correctness of the baselines. In Sec. 5.1, we explain, “as Table 2 confirms, our method and baselines are able to imitate the demonstrations when policies are evaluated in the same task setting as the expert. All methods are able to achieve a near 100% success rate and low distance to goal”. The same holds for the point mass task where baselines achieve near-perfect performance on the train distribution (top row in Table 1). Furthermore, the generalization results in Tables 3 and 4 show baselines can succeed in easier generalization settings but gradually fail as generalization becomes harder.
>
> We also included the source code for the baselines in the submission for reproducibility.

---

> > ### Comment · Reviewer_9FVR · 2022-11-14
> > **Thanks for the clarification**
> >
> > I would just strongly advise the authors to really clarify in the paper that AIRL is failing because it is too slow to learn for 5M steps, not because it can't solve the task (which is what we think looking at figure 2).
> >
> > Besides, I was convinced by the explanation and the additive experiments. I increased my score.

---

> > > ### Author Response · Authors · 2022-11-15
> > > **Response to Reviewer 9FVR**
> > >
> > > We thank the reviewer for the response. We updated the paper to clarify why AIRL fails (changes in red). In addition to changing the language, we added a new paragraph at the end of Sec. 4 to discuss this.

---

### Official Review · Reviewer_RnMY · 2022-10-26

**Confidence:** 4
**Correctness:** 4
**Technical Novelty And Significance:** 4
**Empirical Novelty And Significance:** 4
**Recommendation:** 8

**Clarity, Quality, Novelty And Reproducibility:**

This paper has high clarity and quality.
All text and graphics are well-polished and structured.
Code, hyperparameters, and experimental details are provided for reproduction.

**Strength And Weaknesses:**

### Strength
- The proposed method meaningfully improves the performance of previous imitation learning methods.
Both the generalization quality and robustness of learned policy are consistently outperforming prior imitation learning methods.
- The qualitative results clearly depict the benefit of the method and how it can improve generalization power.
- Paper is well written and easy to follow.

### Weakness
- As clearly described in the paper, BC-IRL should be used with a backbone RL algorithm that can update policy using reward parameters.
This limitation will exclude a meaningful portion of conventional RL algorithms.
- Evaluations are done in a relatively short-horizon and less diverse set of the domain. Few more results on other domains can strengthen the experimental support.

**Summary Of The Paper:**

The submission proposes a novel IRL algorithm.
Instead of prior methods optimizing maximum entropy objective or adversarial objective for reward learning, the proposed method (BC-IRL) seeks reward function that makes a policy updated from the reward be close to expert behavior.
To optimize such an objective, the few-step updated policy parameters are bi-level optimized with respect to reward parameters.
Learned reward function shows improved generalization ability than existing IRL methods and thus it can able to train more robust policy.

**Summary Of The Review:**

This paper is very clearly written and the proposed method is strongly supported by experimental results.
There is a methodological limitation but it doesn't outweigh the benefit of sharing this idea with the community.
Thus, I vote to accept this paper, and I'd be convinced stronger if I can see results in a few more challenging evaluation domains.

---

> ### Author Response · Authors · 2022-11-12
> **Response to Reviewer RnMY**
>
> We thank the reviewer for their comments.
>
> **1. As clearly described in the paper, BC-IRL should be used with a backbone RL algorithm that can update policy using reward parameters. This limitation will exclude a meaningful portion of conventional RL algorithms.**
>
> BC-IRL is compatible with any policy update that is differentiable with respect to the reward function parameters. This includes popular on-policy methods such as PPO (shown in the paper), TRPO, or REINFORCE. This also includes model-based methods like differentiable model predictive control (we investigate this in Sec. A.5). Finally, BC-IRL can be extended to include off-policy algorithms such as Soft Actor Critic (SAC) by differentiating through the Q-function to update the reward.

---

### Decision · Program_Chairs · 2023-01-20

**Decision:**

Accept: notable-top-25%

**Justification For Why Not Higher Score:**

* Limited experimental results on a few environments
* Questionably low results for some of the baselines

**Justification For Why Not Lower Score:**

* Really novel technique for IRL based on bi-level optimization
* The proposed technique mitigates the overfitting issue experienced by many other inverse RL techniques
* Meaningful improvement iover state of the art RL
* Paper is well written and easy to follow.

**Metareview: Summary, Strengths And Weaknesses:**

The paper describes a new technique based on bi-level optimization for inverse RL.

Strengths:
* Really novel technique for IRL based on bi-level optimization
* The proposed technique mitigates the overfitting issue experienced by many other inverse RL techniques
* Meaningful improvement iover state of the art RL
* Paper is well written and easy to follow.

Weaknesses:
* Limited experimental results on a few environments
* Questionably low results for some of the baselines

Overall, this is very interesting work that makes an important contribution to the state of the art of inverse RL with an innovative approach that mitigates overfitting that plagues previous inverse RL techniques.

**Note From Pc:**

if the above contains the word "oral" or "spotlight" please see: "oral" presentation means -> notable-top-5% and "spotlight" means -> notable-top-25%. As stated in our emails, we are disassociating presentation type from AC recommendations